# Practical Challenges in the Diagnosis of SARS-CoV-2 Infection in Children

**DOI:** 10.3390/nursrep15060196

**Published:** 2025-05-30

**Authors:** Alina Petronela Bouari-Coblișan, Claudia Felicia Pop, Valentina Sas, Adina Georgiana Borcău, Teodora Irina Bonci, Paraschiva Cherecheș-Panța

**Affiliations:** 1General Nursing Discipline, Faculty of Nursing and Health Sciences, Iuliu Hațieganu University of Medicine and Pharmacy, 400124 Cluj-Napoca, Romania; petronela.coblisan@umfcluj.ro (A.P.B.-C.); felicia.pop@umfcluj.ro (C.F.P.); 23rd Pediatric Clinic, Children’s Emergency Clinical Hospital, 400315 Cluj-Napoca, Romania; pusachereches@umfcluj.ro; 3Third Pediatric Discipline, Department 8, Mother and Child, Faculty of Medicine, Iuliu Hațieganu University of Medicine and Pharmacy, 400124 Cluj-Napoca, Romania; 4Ophthalmology Network Dr. Holhoș, 510009 Alba Iulia, Romania; 5Pathophysiology, Department of Morphofunctional Sciences, Faculty of Medicine, Iuliu Hațieganu University of Medicine and Pharmacy, 400012 Cluj-Napoca, Romania; adam.teodora@umfcluj.ro

**Keywords:** SARS-CoV-2 testing, nasopharyngeal swab techniques, nurse training, pediatric healthcare workers

## Abstract

**Background/Objectives:** The COVID-19 pandemic, caused by SARS-CoV-2, required the rapid development of diagnostic tests. SARS-CoV-2, part of the betacoronavirus genus, shares characteristics with SARS-CoV-1, including its ability to survive on surfaces, facilitating the spread of the infection. This study analyzes the technique of nasopharyngeal secretion collection for SARS-CoV-2 diagnosis and compares the accuracy of rapid antigen and molecular tests. **Methods**: This study had two components: study A assessed the healthcare personnel training in collecting nasopharyngeal secretions and the discomfort associated with applying a questionnaire. Study B compared rapid antigen test accuracy with RT-PCR among children, through a retrospective analysis. The data were statistically analyzed to assess compliance with the testing protocols. **Results**: In study A, 88 healthcare workers achieved an average compliance score of 7.60 out of 10 regarding the collection procedure. Over 70% of participants correctly followed the fundamental steps of the procedure. Many patients who underwent sample collection reported pain and symptoms such as coughing or sneezing. In study B, 198 pediatric patients were tested using rapid antigen tests, collected simultaneously with RT-PCR. The rapid tests showed a 50% sensitivity and 97.5% specificity. **Conclusions**: This study indicates that nasopharyngeal specimen collection techniques are based on international recommendations, but improvements could be made to reduce discomfort. Rapid antigen tests are helpful for screening due to their high specificity and negative predictive value. Continuous healthcare personnel training and the monitoring of diagnostic techniques remain essential in managing SARS-CoV-2 and other viral infections.

## 1. Introduction

The beginning of the current decade was marked by one of the biggest catastrophes in the medical system that affected the population worldwide. Infection with the novel severe acute respiratory syndrome coronavirus 2 (SARS-CoV-2) has led to a global health crisis [1,2]. The SARS-CoV-2 belongs to the genus betacoronavirus from the family *Coronaviridae*, subfamily *Orthocoronavirinae*. It is a virus with a single-stranded, nonsegmented RNA genome that contains four major structural proteins: the nucleocapsid (N) that surrounds the virion, the transmembrane (M) glycoprotein, the envelope (E) protein, and the spike (S) glycoprotein [3,4,5,6,7].

Before this century, coronaviruses caused mild respiratory tract infections in humans. At the beginning of 2000, severe cases of pneumonia with a high mortality rate occurred in the Guangdong province of China and then spread internationally to Hong Kong, Vietnam, Singapore, Canada, and up to 29 countries across six continents [8]. The disease was subsequently labelled severe acute respiratory syndrome (SARS). After 10 years, another outbreak occurred, caused by the Middle East respiratory syndrome coronavirus (MERS-CoV). It was also of zoonotic origin and caused severe respiratory disease and high mortality up to 34.4%. It only affected people from the countries in or near the Arabian Peninsula, and it was one of the most devastating viruses known to humans [8].

Since December 2019, the entire medical community has been conducting extensive research regarding the epidemiology, transmission, and pathogenicity of coronaviruses [8]. Both SARS-CoV-2 and SARS-CoV-1 are viruses that have similar stability and relatively long viability, being detectable in aerosols up to 3 h after aerosolization [6,9,10]. The presence of these viruses for one day on cardboard and up to 2 to 3 days on plastic and stainless steel might explain the high rate of the spread of the infection. Based on current knowledge, the wide distribution of coronaviruses and the high genetic diversity with the possibility of frequent recombination of their genomes, infectious disease experts estimate that new coronaviruses are likely to appear periodically in humans [6].

Therefore, the virus’s structural characteristics, surface stability, and similarity to other betacoronaviruses justify the need for a rigorous sampling technique and its evaluation in practice—the central objective of our study.

The coronavirus disease 2019 (COVID-19) pandemic highlighted the need for accurate diagnostic tests to manage infectious diseases effectively [11,12,13,14]. Initially, diagnostic tests approved by international medical forums had to be performed by trained personnel and required processing in specially designated laboratories with the highest level of safety [14]. For an accurate diagnosis, strict adherence to collection instructions was necessary to obtain samples from which the three different nucleocapsid proteins in the viral genome could be detected.

The rapid development of SARS-CoV-2 diagnostic tests and their global implementation was possible due to the knowledge gained in the SARS and MERS epidemics, and the transparency in the rapid sharing of SARS-CoV-2 genetics [8].

Accordingly, we included both molecular tests (such as RT-PCR) and antigen-based tests (rapid tests), while also emphasizing that prior experience gained during the SARS and MERS outbreaks played a crucial role in the rapid identification of the viral sequence, the determination of target genes for molecular detection, and the development of sampling and diagnostic protocols. This prior experience was essential for identifying molecular targets and designing sampling and diagnostic protocols relevant to the early response to COVID-19.

The diagnostic accuracy of SARS-CoV-2 infection first requires a correct technique for collecting biological samples. The method of sampling nasopharyngeal secretions was used before the outbreak of the pandemic, mainly by the medical staff from the Otorhinolaryngology departments. During past years, the procedure was commonly used to evaluate patients, both children and adults, with suspected respiratory infections caused by other viruses and some bacteria [15].

In early March 2020, within the first days after the SARS-CoV2 pandemic was declared in Romania, in the Children’s Emergency Clinical Hospital (CECH) Cluj-Napoca, a protocol for the etiological assessment of respiratory infections in children and medical staff was implemented. The protocol included specific sampling, transport, analysis, and requirements for results confirmation. For sampling nasopharyngeal secretions, the Centers for Disease Control and Prevention (CDC) and World Health Organization (WHO) recommendations were adjusted during the pandemic. Nasopharyngeal swabs needed to be specifically manufactured to have long, flexible shafts made of plastic or metal. All nurses, technicians, and physicians involved in specimen sampling received additional training. The primary purpose was the strict requirement to wear personal protective equipment (PPE), including a gown, non-sterile gloves, protective mask, and face shield. Sampling began after the patient removed excess secretions from the nasal passages using a handkerchief [11,16].

The correct procedure involved flexing the head to follow the anatomical trajectory of the nasal passages, lifting the tip of the nose for better visibility, inserting the swab up to the level of the posterior wall of the nasopharynx, holding the swab for a few seconds to absorb secretions, and then gently withdrawing it [15,16,17]. After collection, the swab should be placed in a collection tube, in a closed labeled bag, and transported to the specialized laboratory within 30 min after sampling. The WHO developed several protocols for the detection of SARS-CoV-2 RNA by the real-time polymerase chain reaction (RT-PCR) method [17].

As the SARS-CoV2 pandemic continued and the number of infections increased despite severe measures to limit exposure, techniques were developed to identify the virus quickly. Rapid tests generally have a lower sensitivity, but they allow the identification and early isolation of infected persons, thus avoiding the spread of the virus [11]. In addition to the molecular assays and rapid antigen testing, the infection with SARS-CoV-2 might be diagnosed by testing serum IgM and IgG antibodies, which occur 5 days after the onset of disease.

In all pediatric hospitals in Romania and beyond, the patient care activity was disrupted by the COVID-19 pandemic. The Children’s Emergency Clinical Hospital Cluj-Napoca is a major university hospital in Romania, with many departments. In addition to the Clinical Hospital for Infectious Diseases, patients screened in the emergency department and confirmed with COVID-19 were isolated in specific hospital departments. Our department, the Pediatric Clinic, received patients that had a negative RT-PCR test. As the number of cases and the rate of presentation to the emergency department increased, after the introduction of rapid SARS-CoV-2 tests, patients were admitted in our department if they had a negative rapid test, until the RT-PCR result was available. The entire medical staff in these departments was trained and then evaluated for the correct application of the sampling protocol.

While the diagnostic performance of SARS-CoV-2 tests has been extensively studied, fewer studies have addressed the patient experience, procedural factors, and real-world discrepancies in pediatric settings. This study seeks to evaluate these aspects.

The proper collection of nasal secretions is a critical step in ensuring the accuracy of SARS-CoV-2 testing; however, in real-world pediatric practice, the technique can vary considerably depending on the healthcare provider’s level of experience. In addition, rapid antigen tests—widely used for clinical triage—raise concerns regarding sensitivity, particularly in the early stages of infection or in patients with mild symptoms. Although these methods are broadly applied, studies that assess both the correctness of the sampling technique and the outcomes of rapid testing about RT-PCR in pediatric settings remain relevant and necessary, given the practical variability and clinical implications.

We conducted a study with two primary aims. The first was to assess the training level of nurses and other medical staff from the CECH, Cluj-Napoca, Romania, for the nasopharyngeal secretion sampling technique and their perception of the potential downsides of the procedure. The second objective was to analyze the results obtained through the rapid antigen test and RT-PCR in pediatric patients, based on simultaneous testing conducted in clinical practice.

### Research Questions

The following research questions guided this study:What is the level of preparedness of healthcare personnel for correctly applying the nasopharyngeal sampling technique by recommended protocols?What are the most frequent errors or protocol deviations observed in applying the sampling technique?How is the nasopharyngeal sampling procedure perceived in terms of discomfort or adverse reactions, and how does this perception vary by age, gender, or professional background?What is the level of agreement between the rapid antigen test results and those of RT-PCR in pediatric patients tested concurrently?

## 2. Materials and Methods

### 2.1. Study Design

This research consisted of two studies:

Study A was a cross-sectional observational and descriptive study performed in the 1st, 2nd, and 3rd Pediatric Clinics of the CECH, Cluj-Napoca, Romania, that addressed the medical staff from the CECH between 1 June and 1 July 2021.

Study B was retrospective research, with data collected from the medical files of patients admitted to the 3rd Pediatric Clinic of the CECH Cluj-Napoca, Romania, between 1 February and 30 June 2021. These patients were tested for SARS-CoV-2 infection by RT-PCR and rapid antigenic test. We used the rapid antigen test kits from DDS Diagnostic (Bucharest, Romania), which were approved by the Romanian Health Ministry and CECH Cluj-Napoca, Romania.

### 2.2. Study A: Assessment of Medical Staff Training

#### 2.2.1. Study Objectives

The primary objective of study A was to evaluate the level of training of CECH medical staff for the nasopharyngeal secretion sampling technique. A secondary aim of this study group was to examine healthcare providers’ perceptions regarding the potential downside of the procedure. We elaborated a questionnaire based on the literature recommendations and hospital protocol for these purposes.

#### 2.2.2. Questionnaire Development

A questionnaire was developed based on the literature recommendations and hospital protocols. The questionnaire header specified that the data were collected anonymously and voluntarily, and by completing it, the participants agreed to participate in this study.

#### 2.2.3. Questionnaire Structure

The questionnaire contained 19 questions, structured into three sections:Demographic and professional information

The first seven questions refer to the demographic and personal data of the participants: age, gender, school completed, position in the hospital, years of experience, the CECH department where they work, and where the biological sample was collected for SARS-CoV-2 infection testing. The questions were both open-ended and multiple-choice.
2.Evaluation of sampling technique

The second part of the questionnaire included 10 questions that assessed the sampling technique. The questionnaire focused on key elements of the sampling technique, including patient and staff positioning, identification of the functional nostril, the angle and depth of swab insertion, duration of mucosal contact, proper handling of the swab such as rotation and slow withdrawal, and adherence to nasal hygiene protocols.

Nasal hygiene was defined as blowing the nose before sample collection, but only in cases where the participant presented with rhinorrhea. Accordingly, the question posed to the staff was formulated conditionally: “Was the hygiene of the nasal cavity performed before the test?” (Question 5), which corresponds to, “In case of rhinorrhea, was the nose blown before testing?” The available answer options were as follows:Yes—nasal hygiene (blowing the nose) was performed, implying that the participant had nasal secretions;No—nasal hygiene was not performed, though it remained unclear whether this was due to an omission or because it was not required;This was not the case—the participant did not have rhinorrhea, so this step was not applicable.

These questions had multiple answers, with participants having to choose between “YES”, “NO”, and “I DO NOT KNOW”. For a better interpretation of the answers to the questionnaire, each answer was scored as follows: an answer of “YES” was 1 point, an answer of “NO” was 0 points, and an answer of “I DO NOT KNOW” was 0.5 points. The points obtained for each question were summed, yielding scores between 0 and 10. The maximum score of 10 was achieved when all collection steps for the SARS-CoV-2 test were followed. Total score values between 5 and 6 were considered a poor score, between 6 and 7 a low score, between 7 and 8 a medium score, between 8 and 9 a high score, and between 9 and 10 an excellent score, with the most correct sampling technique. A score between 0 and 4 was considered unsatisfactory, with a faulty technique that does not follow the recommendations.
3.Perception of discomfort and adverse reactions

In Section 3, concerning discomfort or adverse reactions, the questions targeted pain and sensations of sneezing or coughing during the procedure.

The last two questions reflected the discomfort caused by the sampling procedure and were related to the possible adverse reactions experienced by the respondent when they were tested and the nasal secretion was collected for testing.

The study design and flow of participant inclusion, questionnaire structure, and data analysis in study A are summarized in Figure 1.

Study B: Retrospective Analysis of Rapid Antigen Test Reliability

Study objectives

Study B analyzed the results obtained through the rapid antigen test and RT-PCR in detecting SARS-CoV-2 infection in hospitalized children.

#### 2.2.4. Data Collection

Study B analyzed data from the medical records of patients admitted to the 3rd Pediatric Clinic of CECH Cluj-Napoca, Romania, between 1 February and 30 June 2021. These patients were tested for SARS-CoV-2 infection with both the RT-PCR technique and the rapid antigenic test. We included patients under 18 years of age who were tested simultaneously with both tests.

#### 2.2.5. Testing Methods

We used the DDS Diagnosis (Romania) kits for the rapid antigen test, which the Romanian Health Ministry and the CECH Cluj-Napoca, Romania, approved.

The RT-PCR test served as the gold standard for comparison. The rapid antigen test was used as an initial triage tool to facilitate timely isolation and allocation to specific wards, while the RT-PCR test was a confirmatory method. In many cases, both tests were performed within a short time frame to enable prompt clinical decision-making and to ensure diagnostic accuracy. Our department, the Pediatric Clinic, received patients who had a negative RT-PCR test. This approach reflects the specific operational context of our institution during the early stages of the pandemic.

This approach reflects the specific operational context of our institution during the early stages of the pandemic.

The patient testing workflow, inclusion criteria, and structure of study B are presented in Figure 2.

### 2.3. Statistical Analysis

The results were analyzed using the Jamovi Interface, (Version 2.3) [The Jamovi project (2022). Jamovi. Retrieved from [https://www.jamovi.org, (accessed on 06 March 2025)], MedCalc Statistical Software (Version 19.0.3), and Microsoft Excel (Version 16.49).

We calculated positive or negative answers as a percentage of the total questionnaires analyzed in study A. Also, demographic data in both study A and study B were reported as percentages of the total number of participants, with the mean and standard deviation for numerical data.

The compliance with the recommendations for sampling nasal secretions was reported based on the percentage of positive answers and the calculated score according to the methodology presented, based on each answer for the first 10 questions of the questionnaire.

To evaluate differences in correctness scores across hospital departments, we applied the ANOVA test and used the value of *p* < 0.05 for statistical significance. The F-test was used to analyze variance, while the Student’s *t*-test was applied for equal variance comparisons.

When the assumptions were met, the chi-square test was used to analyze qualitative variables. Fischer’s exact test was used when the expected cases were under 5. The Phi coefficient and Cramer’s V were used to assess the strength of association between categorical variables. The Phi coefficient was applied for 2 × 2 contingency tables with dichotomous qualitative variables, while Cramer’s V was used for larger contingency tables.

## 3. Results

### 3.1. Study A

Study A was completed by employees from all three Pediatric Clinics of the CECH Cluj-Napoca, Romania. The questionnaires were submitted to healthcare workers and participants answered voluntarily and returned them to investigators. A total number of 88 questionnaires were filled out entirely and available for analysis.

#### 3.1.1. Demographic and Professional Characteristics

Participants’ average age was 39.1 ± 10.9 years (between 23 and 64 years old), with an average experience in the medical field of 9.8 ± 9.2 years (between 1 and 30 years), and significantly more women than males (78.4% versus 21.6%, *p* = 0.001). Regarding the level of education, thirty-eight (43.2%) of the respondents were physicians who graduated from the Faculty of Medicine, twenty-nine (33.0%) of them were registered nurses who graduated from the Faculty of Nursing, fifteen (17.0%) were nurses with post-secondary education, and six (6.8%) were health care providers with secondary education. The youngest age group included physicians, most of them being young specialists, with an average experience in the medical field of 6.3 ± 8.7 years. Registered nurses had a mean of 13.8 ± 11.9 years of experience working in the hospital, and nurses and gymnasium-trained healthcare providers had a mean experience of 12.3 ± 7.3 years.

#### 3.1.2. Compliance with Sampling Protocol

Compliance with each stage for the correct collection of nasal secretions is summarized in Table 1, questions 1–10.

Adherence to the recommended sampling technique was achieved similarly in the hospital’s different departments, with no significant statistical differences between the correctness scores. The mean scores were 7.47 in the first Pediatric Clinic, 7.43 in the second Pediatric Clinic, and 7.83 in the third Pediatric Clinic (*p* < 0.05).

#### 3.1.3. Most Common Errors in Nasal Secretion Collection

The most frequent deviations from protocol were not lifting the tip of the nose during sampling, incorrect tester positioning (on the side of the nostril being sampled), and failure to correctly identify the functional nose (Table 1).

#### 3.1.4. Presence of Nasal Secretions

Most respondents (73.9%) selected “was not the case”, indicating that most participants did not present with nasal secretions and that nasal hygiene was not required. These cases were not considered protocol deviations. A proportion of 21.6% reported correctly performing nasal hygiene in the presence of rhinorrhea. Only 4.5% answered “No”, but it was impossible to determine whether this reflected an actual omission or a mistaken choice in cases where hygiene was unnecessary. Accordingly, sampling technique adherence was assessed based on the applicability of each procedural step, and the absence of nasal hygiene was considered justified when no secretions were present.

#### 3.1.5. Correctness Score for Nasal Secretion Collection

The average correctness score for nasal secretion collection was 7.60 ± 1.49 (10.2%). A poor score, between 5 and 6, was present in nine of the responders (10.2%), twenty responders (22.7%) scored low, and fourteen persons (15.9%) had a medium score. In comparison, 20 participants (22.7%) presented a high score, and 25 persons (28.4%) had an excellent score, between 9 and 10—none of the nurses or physicians who collected the nasal secretions scored below 5.

#### 3.1.6. Comparison of Scores Between Professional Categories

There were no significant differences between the correctness scores of sampling in the professional categories, with the average scores of 7.41 ± 1.34 in physicians and 7.88 ± 1.68 in nurses (*p* = 0.44). We analyzed the average score of registered nurses and nurses with undergraduate training. There are no significant differences between the techniques of these two categories of nurses (7.86 vs. 7.82, *p* = 0.48).

#### 3.1.7. Adverse Reactions During Sampling

The last two questions refer to the possible adverse reactions experienced by respondents when they were tested and their nasal secretion was collected for testing. Almost half (44.3%) of participants experienced pain during testing, while 81.8% sneezed or coughed during and immediately after the procedure. Only 20.4% reported both pain and sneezing or cough at the same time during sampling.

#### 3.1.8. Factors Influencing Adverse Reactions

We analyzed the correlation between possible individual characteristics and the incidence of side effects reported during sampling (Table 2). Younger participants presented more frequent pain and respiratory or other symptoms due to sampling than participants above 41 years. A higher percentage of females than males experienced either pain or sneezing and cough, but the difference was not statistically significant (*p* = 0.074 and *p* = 0.714).

#### 3.1.9. Influence of Profession and Education Level on Adverse Reactions

The education level and profession were significant factors in our group related to the incidence of sneezing or cough during and after sampling, since nurses and other healthcare providers reported these symptoms more often than physicians (*p* = 0.003). On the other hand, physicians reported a higher incidence of pain during sampling than nurses and other healthcare providers, but the differences were not significant.

#### 3.1.10. Correlation Between Sampling Procedure and Side Effects

We analyzed the correlation between each aspect of the procedure for sampling and the incidence of its side effects, like pain, sneezing, cough, or any other symptoms during the procedure.

#### 3.1.11. Significant Aspects of the Procedure

As shown in Table 3, two aspects of the procedure were statistically significant: slightly bending the head backwards during sample collection (*p* = 0.017) and swab rotation for 10 seconds, which induced sneezing, coughing, or other symptoms (*p* = 0.008).

### 3.2. Study B

Study B evaluated the accuracy of the rapid antigen test for the diagnosis of COVID-19. A total of 198 patients who were admitted to the third Pediatric Clinic were included in the study.
Mean age of the participants: 3.6 ± 4.5 years, ranged between 1 month and 18 years old.Gender distribution: 101 males (51.0%) and 97 females (49%).The majority of the participants came from an urban area (53.0%) (Table 4).

#### Clinical Symptoms at Admission in Study B

Most of our patients in study B presented with fever at admission (67.6%) and cough (39.8%). Almost one third (30.8%) of the patients presented other symptoms like dizziness, poor general appearance, headache, frequent voiding, nasal obstruction, and dysphagia (Figure 3).

Performance of the rapid antigen test, compared to RT-PCR (study B)

Out of the one-hundred and ninety-eight hospitalized patients, only five with a positive RT-PCR test had previously tested negative with the rapid antigen test. Among them, one was an infant, three were preschoolers between 1 and 3 years of age, and one was an adolescent of 14 years of age. Three children above 7 years with a negative RT-PCR test had a positive rapid antigen test. In the remaining 190 patients, both tests for detection of SARS-CoV-2 infection were negative.

Table 5 shows the contingency table of the results obtained from simultaneous testing with the rapid antigen test and RT-PCR, highlighting the distribution of concordant and discordant cases.

As shown in the contingency table, there were no actual positive cases (i.e., no patient tested positive with both the rapid antigen test and RT-PCR). Therefore, the sensitivity and positive predictive value (PPV) could not be calculated. The specificity of the rapid test, using RT-PCR as the reference standard, was 98.45%, while the negative predictive value (NPV) was 97.5%.

## 4. Discussion

This study assessed the training of nurses and physicians for the appropriate procedure for nasopharyngeal sampling for detecting the SARS-CoV-2, as well as the disadvantages and inconveniences during the procedure in tested adults. By the end of 2024, the COVID-19 pandemic remained a worldwide concern, although the European Region showed a decreasing trend of the current circulation of SARS-CoV-2 [17]. From December 2019 to 10 November 2024, the WHO reported over 776.8 million confirmed COVID-19 cases across 234 countries, areas, and territories. The current mortality reached over 7 million cases, the majority of which were during the first 3 years of the outbreak. Although the incidence seems to decrease, in Romania, over 3.5 million people were reported with COVID-19, with 689 new cases during the first week of July 2024 and 158 new cases during the last week of December 2024 [16].

### 4.1. The Assessment of the Technique for Nasopharyngeal Sampling

This procedure was not a frequent practice until the outbreak of the SARS-CoV-2 pandemic. However, the COVID-19 pandemic has led medical personnel to perform nasopharyngeal secretions often [15]. In the pediatric departments, the medical staff was trained according to the existing recommendations in the literature. The present study shows that all the aspects of the procedure, starting with flexing the head to respect the anatomical trajectory of the nasal passages, raising the tip of the nose for more visibility, and inserting the swab up to the level of the posterior wall of the nasopharynx while keeping the tampon for a few seconds and gently withdrawing it, were followed according to the recommendations of the CDC and WHO [11,16].

Similar correctness scores in different departments of the CECH were found, with no significant statistical differences (*p* < 0.05). The collection technique was followed in most cases, with an average correctness score of 7.60 ± 1.49. This result proves that the hospital employees respected the recommendations provided during the training for the procedure. We also evaluated other steps of the procedure, like the correct position of the nurse or physician on the side of the nostril from which the nasopharyngeal secretion was collected (only 59.1% were positioned correctly), lack of establishing the functional nostril before testing in 55.6% of situations, and not lifting the tip of the nose in 42.0% of reported procedures.

These mistakes may be attributed to automatic or habitual movements or time pressure in busy clinical environments. We also noted that such errors can negatively affect patient comfort (leading to more painful or difficult sampling) and, to a lesser extent, may impact sample quality without compromising the results’ validity. These findings may help direct future training efforts toward the steps most prone to error, which could, in turn, improve patient comfort.

Other authors noted that the technique for the nasopharyngeal swab procedure for SARS-CoV-2 implies technical pitfalls and any incorrect aspect might be associated with false-negative results [18]. Bending the head back during collection, insertion of the nasal swab up to the level of the posterior wall, rotation for 10 s, and withdrawal slowly without touching the surrounding tissues and the skin were correctly performed. More than 70% of participants confirmed that they applied the correct technique, when asked about all these steps. In two-thirds of the cases, the study participants had no nasal secretions before testing. This report contradicts the high percentage of patients infected with SARS-CoV-2, in which rhinorrhea is present in 60.1% of them [18,19]. However, this result can be explained by the fact that hospitals’ medical and auxiliary staff have been regularly tested on several occasions to detect asymptomatic carriers of the SARS-CoV-2 virus.

### 4.2. The Downsides of Nasopharyngeal Sampling

COVID-19 affected the health workforce disproportionately, since they were the most exposed to the virus [1]. In addition to the adequate personal protective equipment (PPE), frequent testing for SARS-CoV-2 and many changes in the hospital protocols, including quarantine and isolation protocols, contributed to mental and physical challenges which often required psychological interventions.

Healthcare providers, physicians, nurses, and hospital support staff members in our hospital were regularly tested for SARS-CoV-2, both for screening and whenever they experienced symptoms. In our questionnaire, the last two questions were designed to assess the side effects of the harvesting procedure. Of the study participants, 46.6% said they felt pain and 44.3% reported sneezing or cough because of the procedure. In a study published by Li L., 79.6% of study participants stated discomfort during sample collection [20]. The difference between these results might be explained by the fact that we aimed at pain and not discomfort in our questionnaire, and the perception is different for those downsides.

A critical aspect of our research was identifying factors related to the perception of side effects of the sampling procedure. We analyzed the potential role of age, gender, professional level, and technique. The average age of the study participants was 39.1 years, with an average experience in the medical field of 9.8 years. Young physicians and residents with an average experience in the medical field of 6.2 years represented 43.1% of the responders.

Pain after sampling nasopharyngeal secretions was reported in 53.1% of the responders below 40 years of age, and in only 33.3% of older participants. A total of 71.4% of younger responders and 94.9% of those above 41 years reported sneezing, cough, or other symptoms. In similar research, Li W. et al. found that there are no statistical differences in age, marriage, history of anxiety and depression, pain scores, and anxiety scores between different professions, concluding that this procedure can produce some discomfort but not enough to cause anxiety and depression [21].

We analyzed the correlation between gender and side effects during testing for SARS-CoV-2 infection. Although only 42.1% of men reported pain during sampling compared to women (47.8%), the difference was insignificant. Other symptoms were reported in a smaller percentage of men (26.3% vs. 49.2% of women), without statistical significance (*p* = 0.07). Previous studies reported a significantly higher incidence of disadvantages, including pain during medical procedures in women, who appear to be more anxious and nervous than men [21,22]. This conclusion cannot be generalized considering the small sample size and the predominance of women.

Our results lead us to conclude that those with lower professional training (stretcher-bearers, nurses and other healthcare providers) have a lower pain threshold. This is in accordance with different researchers’ reports, according to which high levels of educational training are associated with less pain [23]. On the other hand, sneezing, cough, or any other symptoms during the sample collection were reported in 57.9% of physicians, compared with 95.2% of other healthcare providers. The education gradient has been studied in relationship with general health status for decades [24]. The main possible explanation relies on better health behavior in more highly educated people, with better general health outcomes. One of the central aspects is the assessment of chronic pain, and several studies have revealed a lower prevalence of pain in people with greater levels of education [25,26]. Some authors have proved no clear relationship between education or socioeconomic position and pain [27,28]. This relationship is complex and includes potential mediators and/or confounders such as psychological well-being, employment status, and health behaviors [23,24,29].

Study A evaluated whether compliance with the sampling protocol influences pain perception during testing. Analyzing the rise of the nose tip, the examiner position, and establishing the functional nostril, values for *p* > 0.05 were found. Regarding the association between bending the head back and the pain during the procedure, our results showed that this is significantly related to side effects (*p* = 0.017). The international recommendations also aim to reduce pain during sample collection and highlight that clinicians should be familiar with the anatomy and proper patient testing method [18]. Performing the hygiene of the nasal cavity before the test might also be a factor related to pain occurrence, but the low number of participants that presented rhinorrhea before the test might interfere with the statistical threshold (*p* = 0.057). The insertion of the nasal swab up to the level of the posterior wall, its rotation for 10 s, and slow withdrawal did not influence the occurrence of either pain or sneezing, cough, or other symptoms.

### 4.3. Diagnosis Accuracy of Rapid Antigen Test

Our second objective was to analyze the accuracy of the diagnosis of rapid antigen tests compared with molecular testing in our pediatric population. Rapid tests are used for the diagnosis of other viral (influenzae, respiratory syncytial virus, parainfluenza, rotavirus) or bacterial (Streptococcal spp., Bordetella spp., Mycoplasma pneumoniae) infections [30,31]. There are several relevant advantages when these rapid tests are available and used. The paramount convenience is that it only takes minutes before an etiological diagnosis is available. Consequently, an appropriate management may be recommended for isolation to reduce the transmission rate and the therapeutic approach. The advantages of these tests are that they can be used in all ages, including infants and immunocompromised patients, are available during weekends, and have substantially lower costs both for the result itself and for the management of the patients.

In our study group, we proved a high specificity (97.50%) and negative predictive value (98.48%) of the rapid test for SARS-CoV2. Similar results are reported by other authors that analyzed the performance of bacterial and viral tests, with a specificity of 88.8% and 98.7% NPV [32]. Since the main reason for hospitalization was fever (67.6%) and cough (39.8%), the most frequent diagnoses were acute upper and lower respiratory tract infections. We gained experience during the COVID-19 pandemic, and various panels for etiological assessments of respiratory tract infections are now available, from limited panels (2-in-1 or 3-in-1 for influenza, SARS-CoV-2 and RSV) to multiplex panels, which include both viral and bacterial agents [31]. The accurate detection of the SARS-CoV2 virus remains an important goal today.

A Cochrane review published in October 2024 shows that the RT-PCR testing technique has an average sensitivity of 95.1% (95% CI 91.1% to 97.3%), and an average specificity of 99.7% (95% CI 98.5% to 99.9%) [33]. In the first quarter of 2020, not all countries had a reasonable capacity and sufficient equipment for RT-PCR testing [34]. Attempts have been made to replace RT-PCR testing with IgM and IgG antibody determination for COVID-19 diagnosis. Still, these do not directly detect the presence of SARS-CoV-2, and the results take several hours and have very low sensitivity and specificity. The advantage is that the collection is concurrent with other serological investigations performed during patient evaluation, and they are less expensive and less time-consuming than other diagnostic methods [35].

Some authors have proposed a web-based COVID-19 diagnostic support system based on machine learning techniques, Heg.IA, which uses blood tests and predicts the need for hospitalization [36]. However, these have not come into current use. Currently, the usefulness of rapid tests is undeniable. In the last 2 years, analyses have been carried out that have highlighted significant differences between the sensitivity and specificity of these tests, depending on the manufacturer [10]. In our hospital practice, we used the DDS Diagnosis (Romania) kits for the rapid antigen test, approved by the Romanian Health Ministry and the CECH Cluj-Napoca, Romania. An extensive Cochrane review evaluated existing rapid and molecular tests [37]. One of the conclusions was that direct comparisons of test brands are needed, an argument that supports the value of the analysis proposed in the present study. Differences could also appear due to the context of symptomatic patients or screening purposes.

A recent review reported a sensitivity of 66.7% up to 100% for the rapid test in adults, and a specificity of 93.7% to 100%. Both the positive predictive value and negative predictive value were very high in symptomatic adults [38]. Another aspect highlighted was the failure to follow the manufacturer’s instructions regarding collection, transport, or long intervals between sample collection and testing [37]. The sensitivity and specificity of the rapid test are different at the onset of symptoms compared to testing in the first week or later. The average sensitivity for detecting infection by a rapid test in asymptomatic participants was 58.1%, with a specificity of 98.9%, which are lower values than those in symptomatic individuals [37].

Our study reported the results of rapid antigen and RT-PCR testing performed simultaneously in pediatric patients who presented to the hospital emergency department with acute symptoms.

Although the negative predictive value was high, a zero sensitivity raises concerns about the potential risk of false-negative results, particularly in the early stages of infection, when the viral load may be low. Given these limitations, in pediatric settings, rapid antigen tests may be used as preliminary triage tools. However, in the pandemic context analyzed, RT-PCR testing was necessary to complement rapid tests in symptomatic cases or those with epidemiological risk, in order to minimize the risk of diagnostic omissions.

The rapid etiological assessment of the patients with either respiratory, gastrointestinal, or general symptoms by using rapid tests contributed to a better isolation of positive cases and reduced the spread of the virus. The quick results of rapid tests allow a better treatment regimen, contribute to a guided management, and the detection of a viral etiology is beneficial in avoiding the overprescription of antibiotics [32]. The early initiation of etiological therapy shortens the duration of the disease, the length of hospitalization, reduces patient care costs, and improves prognosis [31].

Risk factors contributing to false-positive or false-negative results of molecular or rapid tests include technical problems with collection, storage, and transport, suboptimal specimen collection, and the interpretation of results [39]. The testing routine developed during the COVID-19 pandemic must be constantly checked for all aspects of quality so that patient health comes first, along with the safety of medical personnel. The high prevalence and high mortality rate require that the correct diagnostic efforts complement the efforts to prevent the disease through vaccination and accurate and prompt therapy [40].

### 4.4. Limitation of the Study

The main limitation of our study is the relatively small number of participants in study A, who were predominantly women, and the imbalance between the sizes of our age groups and education level subgroups. These aspects could interfere with interpreting the related factors of the incidence of sampling side effects.

The use of a self-administered questionnaire is a significant aspect of our study, as it introduces the possibility of reporting bias. It is important to note that reactions may be influenced by individual perceptions, recall errors, or the tendency to provide socially desirable answers.

Additionally, regarding the analysis of testing outcomes, we note that although all patients with positive results had fever at the time of their emergency department evaluation (when the rapid antigen test was performed), **we do not have precise data on the timing of RT-PCR sample collection or the interval between symptom onset and testing**. These limitations could affect the interpretation of the antigen test’s performance and have been acknowledged in the revised manuscript.

Further studies on larger batches could demonstrate the need to follow these steps to make sampling less uncomfortable. Other factors could bias our results regarding the downsides of the procedure. We did not include some possible relevant questions in our questionnaire, like how many times a subject was tested, if the test was positive, or if a subject was tested because of the presence of symptoms or for screening purposes. These questions could provide a better interpretation of the prevalence of side effects.

## 5. Conclusions

Our study proves that the techniques for collecting nasopharyngeal secretions for detecting the SARS-CoV-2 virus are generally well understood and effectively applied. While many participants followed the essential steps of the protocol, our findings also highlight errors identified in its improper application. This indicates that, although the training was generally practical, further improvements are needed to ensure consistent adherence to all recommended steps and to minimize technical errors during sample collection.

We also introduced a correctness score to assess the sampling procedure objectively.

The perception of pain or other side effects during sample collection can differ with age, gender, and professional training.

Although no positive cases were identified in our dataset, the rapid antigen test demonstrated a high specificity and a significant negative predictive value in the studied pediatric population. Due to the absence of accurate positive results, sensitivity could not be calculated; however, the results support the test’s utility for ruling out infection in low-prevalence settings, such as pediatric populations who initially tested negative through rapid screening.

## Figures and Tables

**Figure 1 nursrep-15-00196-f001:**
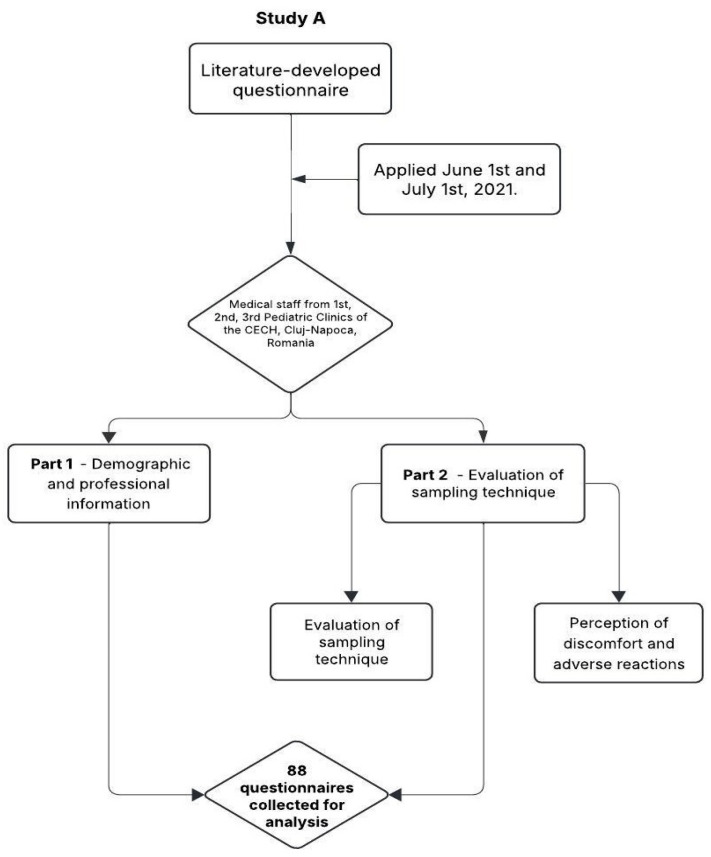
Study A flow chart: design and data collection (N = 88).

**Figure 2 nursrep-15-00196-f002:**
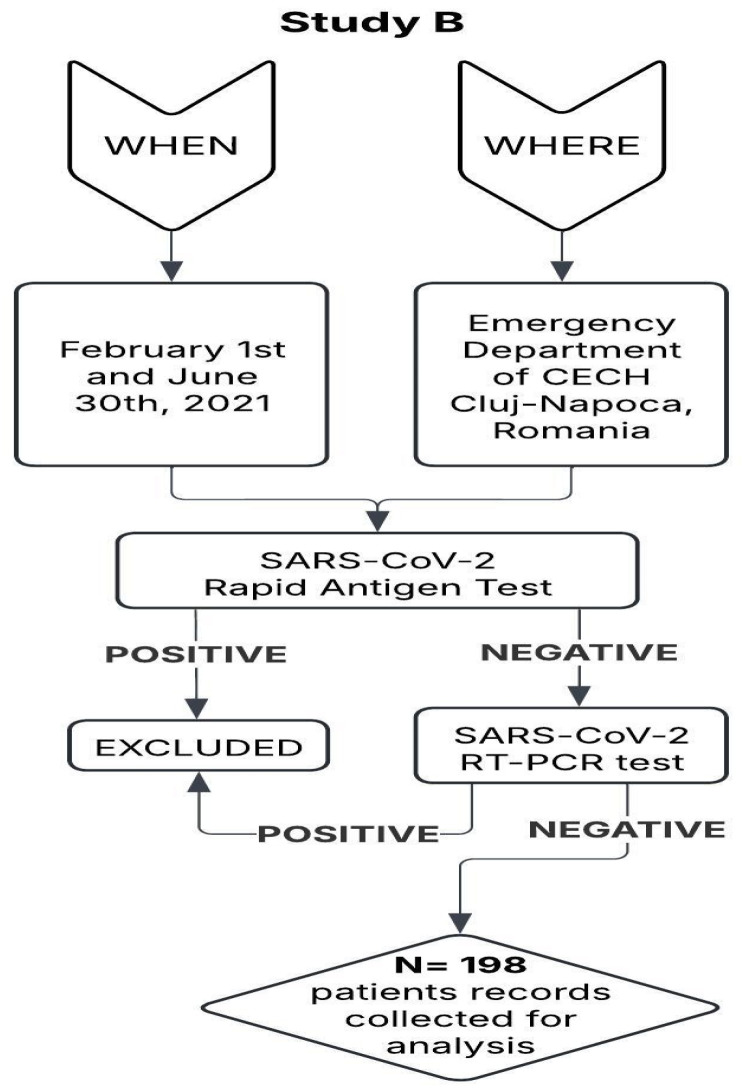
Study B flow chart: testing method and patient inclusion (N = 198).

**Figure 3 nursrep-15-00196-f003:**
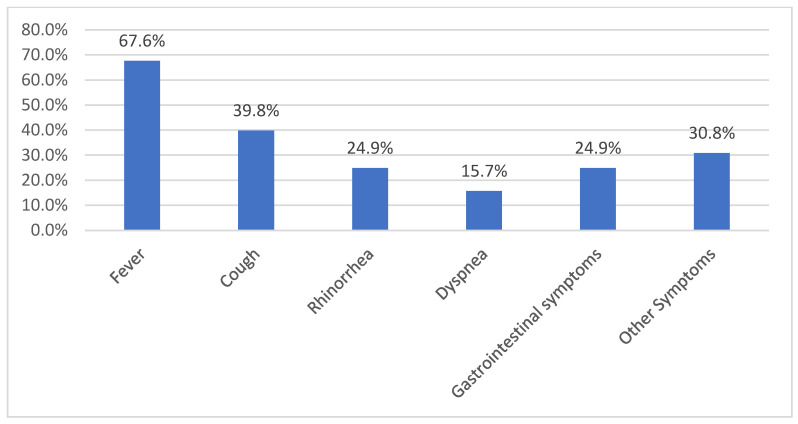
Symptoms at admission.

**Table 1 nursrep-15-00196-t001:** Assessment of the technique of sampling and the perception of the participants on the procedure.

Question	Answer		
	Yes (%)	No (%)	Do Not Know (%)
Was the head slightly bent back during sample collection?	78.4	21.6	
Was the tip of the nose lifted?	52.3	42.0	5.7
Was the nurse positioned on the side of the nostril where the collection was performed?	59.1	37.5	3.4
Was the functional nostril established?	41.0	55.6	3.4
Was the hygiene of the nasal cavity performed before the test?	21.6	4.50	73.9 (was not the case)
Was the nasal swab inserted up to the level of the posterior wall?	93.1	2.30	4.6
Was the pad rotated for 10 s?	77.2	11.4	11.4
Was the nasal swab withdrawn slowly without touching the skin?	86.4	9.10	4.5
Was a mask worn and protective equipment used by the person performing the sample collection?	61.4	37.5	1.1
Was the result of the rapid test communicated within 15 min?	97.8	1.10	1.1
Did you feel any pain during collection of nasal secretion?	46.6	53.4	
Did you sneeze, cough, or have other symptoms during the sample collection?	44.3	55.7	

**Table 2 nursrep-15-00196-t002:** Factors associated with the incidence of side effects of sampling.

	Number	Percentage	*p* Value
**Q11. Did you feel any pain during the nasal swab collection?**			
**Age, years**			0.064
<41 years	26/49	53.1%	
≥41 years	13/39	33.3%	
**Gender**			0.074
Male	5/19	26.3%	
Female	34/69	49.3%	
**Profession, level of training**			0.466
physician	19/38	50.0%	
nurse and registered nurse	13/29	44.8%	
other healthcare provider	7/21	33.3%	
**Q12. Did you sneeze, cough, or have other symptoms during the sample collection?**			
**Age, years**			0.005
<41 years	35/49	71.4%	
≥41 years	37/39	94.9%	
**Gender**			0.714
males	15/19	78.9%	
females	57/69	82.6%	
**Profession, level of training**			0.003
physician	25/38	65.8%	
nurse and registered nurse	22/29	93.1%	
other healthcare provider	20/21	95.2%	

**Table 3 nursrep-15-00196-t003:** Correlation between the procedure and the incidence of side effects.

	Q11. Pain During Procedure			Q12. Sneeze, Cough, or Other Symptoms		
	% Answer			% Answer		
	YES versus NO	*p*	phi/V	YES Versus NO	*p*	phi/V
The head was slightly bent back during sample collection	36.7% 68.4%	0.017	Phi = 0.255	78.3% 94.7%	0.176 *	Phi = 0.17
Tip of the nose lifted	37.0% 52.4%	0.146	Phi = 0.155	78.3% 85.7%	0.365	Phi = 0.09
The nurse on the side of the nostril where the collection was performed	42.3% 47.2%	0.640 *	Phi = 0.0486	82.7% 80.6%	0.790	Phi = 0.027
The functional nostril established	50.0% 40.4%	0.372	Phi = 0.090	88.9% 76.9%	0.152	Phi = 0.15
The hygiene of the nasal cavity before the test **	63.2% 75.0% 36.9%	0.057	V = 0.250	68.4% 75.0% 86.2%	0.114 *	V = 0.192
The nasal swab inserted up to the level of the posterior wall	45.1% 33.0%	0.689	Phi = 0.050	80.5% 100%	0.587 *	Phi = 0.12
The pad rotated for 10 s	47.1% 35.0%	0.340	Phi = 0.102	88.2% 60.0%	0.008 *	Phi = 0.307
The nasal swab was withdrawn slowly	44.7% 41.7%	0.842	Phi = 0.020	82.9% 75.0%	0.451 *	Phi = 0.07

* Fischer exact test (chi-square test assumptions were not met) ** Yes/no/was not the case.

**Table 4 nursrep-15-00196-t004:** Sociodemographic characteristics of patients from study B.

Parameter	Value
Age, mean ± SD	3.6 ± 4.5 years
Age groups, no (%)	
infants, 0–12 months old	58 patients (29.3%)
1–3 years old	83 patients (41.9%)
4–6 years old	20 patients (10.1%)
>7 years old	37 patients (18.7%)
Males, no (%)	101/198 (51.0%)
Environment, no (%)	
urban area	105/198 (53.0%)
rural area	78/198 (39.4%)
not specified	15/198 (7.6%)

**Table 5 nursrep-15-00196-t005:** Contingency table with totals: rapid test vs. RT-PCR.

	RT-PCR Positive	RT-PCR Negative	Total
Rapid Test Positive	0	3	3
Rapid Test Negative	5	190	195
Total	5	193	198

## Data Availability

The data presented in this study are available on request from the corresponding author.

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
