# Peer review of "Practical Challenges in the Diagnosis of SARS-CoV-2 Infection in Children"

_nursrep, 2025, doi:10.3390/nursrep15060196_

Round 1

Reviewer 1 Report

Comments and Suggestions for Authors

The authors aimed to analyzes the technique of nasopharyngeal secretion collection for SARS-CoV-2 diagnosis and compares the effi-ciency of rapid antigen and molecular test.

The study is very interesting.

Introduction is very well written.

Methods are robust.

Results are relevant, as the authors found that nasopharyngeal specimen collection techniques are properly based on international recommendations, although improvements could be made to reduce discomfort. Rapid antigen assessment is helpful for screening due to high specificity and negative predictive value. Continuous healthcare personnel training and monitoring of diagnostic techniques is pivotal in managing SARS-CoV-2 and other viral infections.

Discussion deepens all the issues related to the study.

Limitations are well-acknowledged.

I accept the paper as it is, without modifications.

Author Response

We thank Reviewer 1 for their time and positive assessment. We appreciate the careful reading of our manuscript.

Reviewer 2 Report

Comments and Suggestions for Authors

I have carefully reviewed the manuscript "Practical challenges in the diagnosis of SARS-CoV-2 infection 2 in children". This paper is interesting, but several issues need to be addressed before publication. My detailed comments are as follows:

  1. Please emphasize the rationale of the study in the introduction section, however, sufficient studies have been conducted to validate and compare the diagnostic tests for COVID-19.
  2. You wrote on page 4, line 177, "We included patients between 0 and 18 years old". However, the children should be under the age of 18 years. Please correct accordingly.
  3. Use the term physician instead of "doctor" throughout the manuscript.
  4. "Did you sneeze, cough, or have other symptoms during the sample collection"? Other symptoms should be detailed in the methods section.
  5. It is interesting to check the reliability of the questions in the questionnaire and calculate Cronbach's alpha and construct variables from the questions using factor analysis before the results that you present in the results section in study A.
  6. You wrote on Table 2, "the two age categories were below 40 years and above 41 years". It should be modified to include the ages 40 and 41 years, such as <41 vs. ≥41
  7. You wrote on page 9, lines 300-306, "Out of the 198 hospitalized patients, only 5 had a positive RT-PCR test and previously tested negative with the rapid antigenic test. One patient with a positive RT-PCR test was an infant, three were preschoolers between 1 and 3 years of age, and one was a boy of 14 years of age. Three children above 7 years with a negative RT-PCR test had a positive rapid antigenic test. In the remaining 190 patients, both tests for detection of SARS-CoV-2 infection were negative". According to this data, the true positive is zero (zero patients have positive results in both tests for detection of SARS-CoV-2 infection), thus, you need to clarify how to calculate the sensitivity and PPV. However, the specificity should be 98.45% (190/193) and the NPV 97.5% (190/195). It is better to organize the data in a table.
  8. There is no mention of reporting bias due to the self-completion of the questionnaire in the study limitations.
  9. A sound principle is that the conclusions of a scientific article should be grounded in the findings presented in the article. This study cannot conclude that "the techniques for collecting nasopharyngeal secretions for detecting the SARS-CoV-2 virus comply with international recommendations, and the examiner is well-trained and respects the technique protocol". However, you wrote that the rapid antigenic test is highly sensitive, but in the result section, the sensitivity of the test was 50%?

Author Response

Author's Reply to the Review Report (Reviewer 2)

Comments 1: [Please emphasize the rationale of the study in the introduction section, however, sufficient studies have been conducted to validate and compare the diagnostic tests for COVID-19.]

Response 1: We understood the importance of clarifying that the rationale of our study is not to validate or compare diagnostic tests but rather to explore how they are applied in pediatric clinical practice. Therefore, we revised the text to reflect this perspective better. We fully agree with this comment and appreciate the reviewer’s suggestion. Thus, we added the following sentence to the introduction to clarify the study rationale: [While the diagnostic performance of SARS-CoV-2 tests has been extensively studied, fewer studies have addressed the patient experience, procedural factors, and real-world challenges specific to pediatric settings.] This change is marked in red for clarity on page 3, paragraph 4, lines 131, 132, and 133 of the revised manuscript.

We also revised the wording of the second study objective to avoid implying a comparative diagnostic accuracy analysis and instead emphases the real-life clinical testing context. The updated text, now included at the end of the introduction, reads: [The second objective was to analyse the results obtained through the rapid antigen test and RT-PCR in pediatric patients, based on simultaneous testing conducted in clinical practice.] [This change is marked in red for clarity on page 3, paragraph 5, lines 146, 147, and 148 of the revised manuscript.] 

Comments 2: [You wrote on page 4, line 177, "We included patients between 0 and 18 years old". However, the children should be under the age of 18 years. Please correct accordingly.]

Response 2: We have revised the sentence to: [We included patients under 18 years of age.] Thank you for your feedback. [This change is marked in red for clarity on page 4, paragraph 4, line 181 of the revised manuscript.]

Comments 3: [Use the term physician instead of "doctor" throughout the manuscript.]

Response 3:  [I have replaced all instances of the term doctor with physician throughout the manuscript.] Thank you for pointing this out. I agree with the recommendation. [These changes can be found on the following pages: page 3, paragraph 1 and lines 100; page 8, paragraph 4 and lines 290 and 294; page 9, paragraph 1 and line 327 and  page 9, paragraph 2 and line 331; page 7, paragraph 1 and line 274; page 10, paragraph 2 and line 351 and 352; page 13, paragraph 3 and line 433; page 14 paragraph 3 and line 460, page 14 paragraph 4 and line 472, page 14 paragraph 7 and line 494.] [The updated wording in the revised version of the manuscript has been marked in red.]

Comments 4: ["Did you sneeze, cough, or have other symptoms during the sample collection"? Other symptoms should be detailed in the methods section.]

Response 4: [We retained the original question formulation, which included a prompt for participants to report any 'other symptoms'. However, only sneezing and coughing were mentioned in the responses, leading us to not include further details in the methods section. In future studies, we are committed to refining this item to ensure a more specific assessment of potential reactions, providing a more comprehensive understanding of the issue.] Thank you for your comment.

Comments 5: [It is interesting to check the reliability of the questions in the questionnaire and calculate Cronbach's alpha, and construct variables from the questions using factor analysis before the results that you present in the results section in study A.]

Response 5: [We acknowledge the importance of checking the internal consistency of questionnaire-based items and conducting exploratory analyses such as factor analysis to construct latent variables. However, in the current version of the study, we focused on presenting descriptive and directly interpretable results based on the structure of the original questionnaire. Given the limited sample size and the exploratory nature of Study A, we opted not to include Cronbach’s alpha or factor analysis in the analysis. We will consider including such methodological approaches in future studies with larger and more homogeneous samples.] Thank you for this valuable suggestion.

Comments 6: [You wrote on Table 2, "the two age categories were below 40 years and above 41 years". It should be modified to include the ages 40 and 41 years, such as <41 vs. ≥41.]

Response 6: [We have corrected the description of the age categories in Table 2 to reflect the appropriate grouping, now presented as <41 vs. ≥41 years.] Thank you for your observation. [In addition, as part of our own initiative, we bolted the main variables in the table — age, gender (explicitly labeled above the “male” and “female” categories), and level of training — to improve readability and ensure a clearer and more coherent presentation of the analysand data.] [This change was also updated in the main text where this categorization was mentioned and is marked in red.]

Comments 7: [You wrote on page 9, lines 300-306, "Out of the 198 hospitalized patients, only 5 had a positive RT-PCR test and previously tested negative with the rapid antigenic test. One patient with a positive RT-PCR test was an infant, three were preschoolers between 1 and 3 years of age, and one was a boy of 14 years of age. Three children above 7 years with a negative RT-PCR test had a positive rapid antigenic test. In the remaining 190 patients, both tests for detection of SARS-CoV-2 infection were negative". According to this data, the true positive is zero (zero patients have positive results in both tests for detection of SARS-CoV-2 infection), thus, you need to clarify how to calculate the sensitivity and PPV. However, the specificity should be 98.45% (190/193) and the NPV 97.5% (190/195). It is better to organize the data in a table.]

Response 7: We acknowledge the inconsistency regarding the sensitivity and PPV values reported in the earlier version of the manuscript. After a thorough review of the data and in collaboration with a statistician, we confirmed that there were no true positive cases — i.e., no patients tested positive on RT-PCR and rapid antigen tests. While sensitivity and PPV can technically be calculated, the absence of true positives means that these metrics are not meaningful or interpretation in our sample. Therefore, we chose not to report them. Thank you for this detailed and accurate observation. [We have corrected the Results section accordingly and clarified the values of specificity and NPV, which remain interpretation.]

Additionally, we revised the title of the subsection from “Accuracy of the Rapid Test in Study B” to [Performance of the Rapid Antigen Test Compared to RT-PCR (Study B)] for clarity and to better reflect the comparative nature of the analysis, is marked in red, page  12, paragraph 2, line 383.]

[Additionally, we added the following introductory sentence before Table 5 to improve clarity: [Table 5 shows the contingency table of the results obtained from simultaneous testing with the rapid antigen test and RT-PCR, highlighting the distribution of concordant and discordant cases.] These changes can be found on page 12, paragraph 3 and lines 390, 391 and 392, which are marked in red.] [In response to your suggestion, we have reorganized the data into a clear and concise 2x2 contingency table, which is now included in the revised Results section.] We have also included a brief explanatory note below Table 5, stating that [As shown in the contingency table, there were no actual positive cases (i.e., no patient tested positive with both the rapid antigen test and RT-PCR). Therefore, sensitivity and positive predictive value (PPV) could not be calculated. The specificity of the rapid test, using RT-PCR as the reference standard, was 98.45%, while the negative predictive value (NPV) was 97.5%.] [These changes can be found on page 13, paragraph 1, and lines 403, 404, 405, 406 and 407, which are marked in red.]

Comments 8: [There is no mention of reporting bias due to the self-completion of the questionnaire in the study limitations.]

Response 8: [The use of a self-administered questionnaire is a significant aspect of our study, as it introduces the possibility of reporting bias. It's important to note that reactions may be influenced by individual perceptions, recall errors, or the tendency to provide socially desirable answers.]. [Thank you for highlighting this point..] [Consequently, I have made the necessary changes, which can be found marked in red on page 16, paragraph 7, lines 596, 597, 598 and 599, which are marked in red.]

Comments 9: [A sound principle is that the conclusions of a scientific article should be grounded in the findings presented in the article. This study cannot conclude that "the techniques for collecting nasopharyngeal secretions for detecting the SARS-CoV-2 virus comply with international recommendations, and the examiner is well-trained and respects the technique protocol". However, you wrote that the rapid antigenic test is highly sensitive, but in the result section, the sensitivity of the test was 50%?]

Response 9: We fully agree that the conclusions of a scientific article must be grounded in the findings presented in the study. Accordingly, we revised the manuscript's conclusions to reflect the data obtained accurately. [We removed the categorical statement that the sampling technique "complies with international recommendations"] and replaced it with a formulation stating that: [While many participants followed the essential steps of the protocol, our findings also highlight errors identified in its improper application. This indicates that, although the training was generally practical, further improvements are needed to ensure consistent adherence to all recommended steps and to minimize technical errors during sample collection.] Thank you for your valuable observation.  [These changes can be found on the following pages: page 17, paragraph 3 and lines 615, 616, 617, 618 and 619.]  [We have carefully re-evaluated the diagnostic performance data with a statistician. Indeed, based on the available data, there were no actual positive cases (i.e., patients who tested positive with both the rapid antigen test and RT-PCR). Therefore, the sensitivity could not be calculated.] Thank you for this critical observation. [We have updated the Results and Discussion sections to reflect this more accurately.  We now state that sensitivity could not be computed due to the absence of true positives in the analyzed sample. Additionally, we have revised the conclusions to avoid any overstatement regarding the diagnostic value of the rapid antigen test. [Although no positive cases were identified in our data-set, the rapid antigen test demonstrated high specificity and a significant negative predictive value in the studied pediatric population. Due to the absence of accurate positive results, sensitivity could not be calculated; however, the results support the test's utility for ruling out infection in low-prevalence settings, such as pediatric populations who initially tested negative through rapid screening.] [These changes can be found on the following pages: page 17, paragraph 6 lines 624, 625, 626, 627, 628 and 629.] [The updated wording in the revised version of the manuscript has been marked in red.]

Reviewer 3 Report

Comments and Suggestions for Authors

The study titled Practical challenges in the diagnosis of SARS-CoV-2 infection in children is a very interesting and well-written article with potential for publication.I have a few minor revision suggestions

  1. Please add some more paragraphs in the introduction section that reinforce the purpose of the study and explain why the study should be written. Present the research questions of your study as a title under this section
  2. The tables in your work are very nice. In the method section, you stated that you carried out the work in 2 prints. However, there is not much clarity and comprehensibility. Therefore, to eliminate this problem, please add a flow chart to your work.     3. There are many results and suggestions in your study. Please expand the conclusion section.

Author Response

Author's Reply to the Review Report (Reviewer 4)

The study titled Practical challenges in diagnosing SARS-CoV-2 infection in children is a fascinating and well-written article with potential for publication.  I have a few minor revision suggestions

Comments 1: [Please add more paragraphs in the introduction section that reinforce the purpose of the study and explain why the study should be written. Present the research questions of your study as a title under this section.]

Response 1: [Proper collection of nasal secretions is a critical step in ensuring the accuracy of SARS-CoV-2 testing; however, in real-world pediatric practice, the technique can vary considerably depending on the healthcare provider’s experience level.] [In addition, rapid antigen tests—widely used for clinical triage—raise concerns regarding sensitivity, particularly in the early stages of infection or in patients with mild symptoms.] [Although these methods are broadly applied, studies that assess the correctness of the sampling technique and the outcomes of rapid testing of RT-PCR in pediatric settings remain relevant and necessary, given the practical variability and clinical implications.] [These changes can be found on page 3, paragraph 5, and lines 136, 137, 138, 139, 140, 141, 142, 143 and 144.]

Research Questions [These changes can be found on page 3, paragraph 7, and lines 151, 152, 153, 154, 155, 156, 157, 158, 159, 160 and 161, which are marked in red.]

The following research questions guided this study:

  1. What is the level of preparedness of healthcare personnel for correctly applying the nasopharyngeal sampling technique by recommended protocols?
  2. What are the most frequent errors or protocol deviations observed in applying the sampling technique?
  3. How is the nasopharyngeal sampling procedure perceived in terms of discomfort or adverse reactions, and how does this perception vary by age, gender, or professional background?
  4. What is the level of agreement between the rapid antigen test results and those of RT-PCR in pediatric patients tested concurrently?

Comments 2: [The tables in your work are very nice. In the method section, you stated that you carried out the work in 2 prints. However, there is not much clarity and comprehensibility. Therefore, to eliminate this problem, please add a flow chart to your work. ]   

Response 2:  [To improve clarity and enhance the reader's understanding of the methodology, we have included two flow charts in the revised manuscript.] [Thank you for your valuable observation.]

Figure 1 illustrates Study A's structure and participant flow (questionnaire-based assessment). [These changes can be found on page 5, paragraph 5 and lines 224 and 225, which are marked in red.]

Figure 2 presents Study B's testing process and inclusion criteria (retrospective analysis).
These additions are intended to summaries the study design and improve overall comprehensibility visually. Both figures have been inserted in the Materials and Methods section and are referenced in the text. [These changes can be found on page 6, paragraph 6, and lines 251, 252, which are marked in red.]

These additions are intended to summaries the study design and improve overall comprehensibility visually. Both figures have been inserted in the Materials and Methods section and are referenced in the text.

Comments 3: [There are many results and suggestions in your study. Please expand the conclusion section.]

Response 3: [In the revised manuscript, we have substantially extended the conclusion to more clearly summaries the design and findings of both Study A and Study B.] We emphasis the main results regarding protocol adherence and training needs identified in Study A, as well as the diagnostic performance and limitations observed in Study B.] [In addition, we highlighted the practical implications of the findings in pediatric care settings and included recommendations for further training and future research directions.] [Thank you for your helpful suggestion to expand the conclusion section.]
[These changes were made to reflect better the study's complexity and potential impact on clinical practice. The revised conclusions are presented on page 16, paragraph 9, lines 620, 621, 622, 623 and 624; page 17, paragraph 3, lines  629, 630, 631, 632, 633 and 634, are marked in red in the updated manuscript.]

Round 2

Reviewer 2 Report

Comments and Suggestions for Authors

The authors have responded to all comments satisfactorily, thus the revised manuscript is suitable for publication.